# Diversity of Studies on Neighborhood Greenspace and Brain Health by Racialized/Ethnic Group and Geographic Region: A Rapid Review

**DOI:** 10.3390/ijerph20095666

**Published:** 2023-04-27

**Authors:** Lilah M. Besser, Marcia Pescador Jimenez, Cameron J. Reimer, Oanh L. Meyer, Diana Mitsova, Kristen M. George, Paris B. Adkins-Jackson, James E. Galvin

**Affiliations:** 1Comprehensive Center for Brain Health, Miller School of Medicine, University of Miami, Miami, FL 33433, USA; 2Department of Epidemiology, School of Public Health, Boston University, Boston, MA 02118, USA; 3Department of Earth & Environment, Boston University, Boston, MA 02118, USA; 4Department of Neurology, School of Medicine, University of California Davis, Sacramento, CA 95817, USA; 5School of Urban and Regional Planning, Florida Atlantic University, Boca Raton, FL 33431, USA; 6Department of Public Health Sciences, School of Medicine, University of California Davis, Davis, CA 95817, USA; 7Departments of Epidemiology and Sociomedical Sciences, Columbia University, New York, NY 10032, USA

**Keywords:** greenspace, brain health, Alzheimer, cognition, race, health disparities

## Abstract

Studies examining associations between greenspace and Alzheimer’s disease and related dementia (ADRD) outcomes are rapidly on the rise, yet no known literature reviews have summarized the racialized/ethnic group and geographic variation of those published studies. This is a significant gap given the known disparities in both greenspace access and ADRD risk between racialized/ethnic groups and between developed versus developing countries. In this rapid literature review, we (1) describe the diversity of published greenspace–brain health studies with respect to racialized/ethnic groups and geographic regions; (2) determine the extent to which published studies have investigated racialized/ethnic group differences in associations; and (3) review methodological issues surrounding studies of racialized/ethnic group disparities in greenspace and brain health associations. Of the 57 papers meeting our inclusion criteria as of 4 March 2022, 21% (*n* = 12) explicitly identified and included individuals who were Black, Hispanic/Latinx, and/or Asian. Twenty-one percent of studies (*n* = 12) were conducted in developing countries (e.g., China, Dominican Republic, Mexico), and 7% (*n* = 4) examined racialized/ethnic group differences in greenspace–brain health associations. None of the studies were framed by health disparities, social/structural determinants of health, or related frameworks, despite the known differences in both greenspace availability/quality and dementia risk by racialized/ethnic group and geography. Studies are needed in developing countries and that directly investigate racialized/ethnic group disparities in greenspace—brain health associations to target and promote health equity.

## 1. Introduction

Alzheimer’s disease (AD) and related dementias (ADRD) are a group of neurodegenerative diseases that significantly impact an individual’s daily life and functional abilities. These conditions typically differ in their clinical presentations depending on etiology. For instance, AD often initially presents with episodic memory deficits (i.e., recollection of personal experience/past events) [1] and Parkinson’s disease with motor decline [2]. ADRD prevalence is projected to increase from the current 6.7 million [3] to over 13 million by 2050 [4]. Concurrently, the population will become more racially/ethnically diverse, with the Black population expected to grow by 34% and the Hispanic/Latinx population expected to grow by 86% in the next three decades [5]. ADRD are responsible for significant physical, social, and economic burdens on patients, families, and the healthcare system [6], and minoritized groups bear the brunt of these burdens as patients and caregivers [7]. Black and Hispanic (versus White) individuals have a higher risk of ADRD (1.5 to 2 times as high) [8]. In addition, health conditions such as cardiovascular disease and diabetes, which are associated with increased ADRD risk, are more prevalent in Black and Hispanic individuals [9,10,11].

ADRD disparities between minoritized, racialized/ethnic groups and between developed and developing countries (e.g., higher versus lower-middle income) are well documented and hypothesized to result from structural and social determinants of health (S/SDOH) [12,13,14,15]. Structural determinants of health are the culture, values, policies, practices, and laws that shape a society (e.g., genderism, racism, capitalism). They are the overarching factors that shape living, working, and educational environments, resources, and opportunities (i.e., social determinants of health) that in turn impact individual-level health. Social determinants of health broadly include healthcare access and quality, educational access and quality, social and community context, neighborhood and built environment, and economic stability [16]. Thus, for example, disparities in S/SDOH, such as fewer educational opportunities, higher rates of poverty, and greater exposure to adversity and discrimination by racialized/ethnic groups, may help explain the greater risk of ADRD in Black and Hispanic communities [17,18].

S/SDOH influence environmental exposures, health behaviors, and quality of life throughout the life course, and these mechanisms are posited to affect brain health. Brain health is “the state of brain functioning across cognitive, sensory, social-emotional, behavioral, and motor domains, allowing a person to realize their full potential over the life course ” [19]. Social determinants are inextricably tied to the modifiable risk factors known to decrease brain health and increase ADRD risk (e.g., social contact, air pollution exposure, physical activity, and depression), and they differ by the racialized/ethnic group [20]. For instance, an analysis reported that worldwide, the highest estimated population-attributable risk (PAR) for ADRD was for low educational attainment (19%), while in the US, the highest estimated PAR was for physical inactivity (21%) [21]. A separate US study found that the modifiable risk factors most prominently associated with ADRDs were midlife obesity, physical inactivity, and low education. The proportion of ADRD cases associated with modifiable risk factors was higher among American Indian and Alaska Native, Black, and Hispanic individuals compared with Asian and White individuals [22].

The social-ecological model posits that community-level and individual-level factors influence health, yet only recently more attention has been directed toward community factors associated with ADRD outcomes [23]. Greenspaces, which are outdoor areas with natural vegetation, including but not limited to parks and gardens, are an integral part of the environment in which people live, learn, work, and age (Figure 1). They can provide physical and mental health benefits, and preliminary evidence suggests that they might help lower the risk of neurodegenerative diseases. A rapid review of 22 studies published as of 13 February 2020, found that 77% of studies demonstrated beneficial greenspace—brain health associations for children through older adults [24], and a meta-analysis of 12 studies on greenness and dementia risk published as of 30 March 2022, found that intermediate levels of greenness were associated with a slightly reduced risk of dementia [25]. Greater greenspace access in childhood, adulthood, and older age has been associated with slower cognitive decline over time in middle- and older-age adults [26,27]. Similarly, greener neighborhoods (more healthy, green vegetation) in childhood, adulthood, and older age have been associated with better cognitive functioning among middle- to older-age adults [26,28,29]. Few studies have focused on ADRD-related biomarkers, such as from magnetic resonance imaging (MRI), but early evidence suggests positive associations between more neighborhood greenspace and better brain imaging outcomes (e.g., amygdala integrity, larger regional brain volumes) in middle- to older-age adults [30,31].

Increasing the quantity and quality of community greenspaces might provide a novel and feasible public health intervention to reduce ADRD risk [32], particularly for historically disadvantaged neighborhoods [33,34]. For example, in the US, census tracts with a higher proportion of minoritized, racialized/ethnic groups had less greenspace in 2001 and lost more greenspace between 2001 and 2011 compared with census tracts with lower proportions of minoritized groups [33]. In addition, studies of several U.S. cities found that minoritized individuals with lower socioeconomic status (SES) living in redlined neighborhoods [34] (i.e., experienced historic discriminatory mortgage lending practices), have less neighborhood tree canopy [35], greenspace [36,37], and parks [38,39], although this is not universally the case [40].

Studies examining associations between greenspace and ADRD outcomes have been rapidly on the rise since a previous rapid review [24] on this topic, yet no known reviews have summarized the racialized/ethnic group and geographic variation of the published studies. This is a significant gap given the known disparities in both greenspace access and ADRD risk between racialized/ethnic groups and between developed and developing countries [12,13,14,34,41]. Thus, in this study, we aim to determine the extent to which published greenspace—brain health studies have examined different racial/ethnic groups and geographic regions. We conducted a rapid literature review [42] to delineate the diversity of published greenspace—brain health studies with respect to included racialized/ethnic groups and geographic regions, the extent that published studies have investigated racialized/ethnic group differences in associations, and methodological issues surrounding studies of racialized/ethnic group disparities in greenspace and brain health associations. The overarching goal of this review is to provide preliminary evidence and expose gaps in the study of underrepresented racialized/ethnic groups and lower- to middle-income/developing countries among greenspace—brain health studies to inform future systematic reviews and primary research.

## 2. Materials and Methods

We build upon the work of a previous rapid literature review [24], employing the same inclusion criteria, databases, and search terms. However, in the current review, we include an additional two years of published papers (2020–2022) and, unlike the prior review, provide a unique focus on the racialized/ethnic group and geographic diversity of the studies. Rapid reviews employ methods similar to systematic or scoping literature reviews, allowing an evaluation of existing literature on emerging topics but differ in that they are typically conducted over a shorter time span by fewer individuals when time and resources are limited [42]. The prior review searched PubMed, Web of Science Core Collection, and Embase for papers published through 13 February 2020 that included greenspace as the exposure and ADRD brain health outcomes, and excluded papers that were (1) not in English; (2) not primary research studies; (3) focused on indoor greenspaces/views or virtual reality views; (4) ecological studies; (5) focused on attention restoration/mental fatigue (short-term effects); or (6) focused on greenspace activities (e.g., gardening). For this review, we automatically included the 22 papers published in the prior rapid review and repeated the search methods and inclusion/exclusion criteria to add any new papers published in the approximately 2 years following that review (i.e., 14 February 2020 to 4 March 2022). A single reviewer (LMB) performed searches of titles and abstracts for the following two sets of keywords (the same keywords searched for in the prior review):“greenspace or green space or greenness or parks or park or park space or parkspace” AND “cognition or cognitive or memory or brain aging or Alzheimer or Alzheimer’s or dementia or cognitive impairment”“neighborhood environment or wilderness or greenery or natural space or natural environment or public garden or recreational resource or normalized difference vegetation index or built environment or open space or woodland” AND “brain volume or brain atrophy or neurodegenerative disease or Alzheimer biomarker or cognition or cognitive or memory or brain aging or Alzheimer or Alzheimer’s or dementia or cognitive impairment”

Papers with titles and abstracts meeting the inclusion criteria moved on to a full-text review by a single reviewer (LMB), and those continuing to meet the criteria contributed to the final sample. Three reviewers (LMB, MPJ, and CR) divided up the final set of papers and charted a predetermined set of fields on (1) basic characteristics of the papers (author, year published, study location, data source, sample size, age groups under study, and statistical methods); (2) details on the specific greenspace and ADRD outcome measures (e.g., objective or self-reported; geographic information systems (GIS) buffers of interest (if applicable); measured at a one-time point or as changes over time); (3) observed associations between greenspace and ADRD outcomes; (4) details regarding racialized/ethnic groups under study, statistical methods for examining racialized/ethnic group differences in associations, and observed associations by the racialized/ethnic group; and (5) authors’ discussions of health disparities framework or related conceptual models, such as social determinants of health or structural racism. The detailed charted data are available in Appendix A.

We created figures to display the number of studies by age group under study, participant racialized/ethnic group, geographic region, greenspace measure, ADRD outcome measure, studied associations (e.g., between greenness and cognition or between park space and ADRD diagnosis), and observed association (positive, negative, or null). The following age groups were chosen based on the importance of different life stages to brain development [43,44] and ADRD risk/pathology development [45]: 0–18 years (hereafter termed childhood for simplicity), 18–44 years (young adulthood), 45–64 years (middle age), and ≥65 years (older age). Only participants who were specifically identified by their racialized/ethnic group in the study were categorized as such in this study, and participants from studies without such specificity were categorized as “not specified”.

For the figure on geographic region, studies from the U.S., Canada, Australia, Western Europe, Lithuania, Bulgaria, and Japan were categorized as developed [46]. The remaining developing country categories included Latin America/Caribbean and East Asia (no studies were from South Africa, the Middle East/North Africa, or South Asia).

## 3. Results

The final sample included 57 papers published through 4 March 2022 (Figure 2) [26,27,29,30,31,47,48,49,50,51,52,53,54,55,56,57,58,59,60,61,62,63,64,65,66,67,68,69,70,71,72,73,74,75,76,77,78,79,80,81,82,83,84,85,86,87,88,89,90,91,92,93,94,95,96,97,98]. Papers that met eligibility criteria followed a sequential review of titles, abstracts, and full text; duplicates across the three databases were eliminated; one study with duplicate reporting of results and one study focused only on school-level cognitive test scores (i.e., ecological study) were removed; and three new papers identified from a review of reference lists of the papers meeting our criteria in the rapid review were added.

The sample size, age groups, geographic locations, racialized/ethnic groups included, and observed associations for each of the 57 reviewed studies are provided in Table 1 and Figure 3A–F. Of note, the figures report non-mutually exclusive percentages because some studies include multiple categories (e.g., multiple age groups or outcomes of interest). By age group, 19 studies included < 18-year-olds (33%), 7 studies included 18–44-year-olds (12%), 20 studies included 45–64-year-olds (35%), and 38 studies included ≥65-year-olds (67%) (Figure 3A). The large majority (*n* = 46, 81%) of the studies were in developed countries, with the remaining conducted in Latin America/Caribbean (*n* = 1, 2%) and East Asia (*n* = 12, 21%) (Figure 3B). Among those studies explicitly identifying the racialized/ethnic group of participants, 10 included Black individuals (18%), 5 included Hispanic/Latinx individuals (9%), 5 included Asian individuals (9%), and 14 included White individuals (25%) (i.e., studies identified at least a subsample of these individuals) (Figure 3C). Altogether 12 studies (21%) specifically mentioned including individuals who were Black, Hispanic/Latinx, or Asian. Seventy-two percent of studies (*n* = 41) did not specify the participants’ racialized/ethnic groups, and as described in the methods, we did not automatically assign racialized/ethnic groups based on a presumptive racialized/ethnic group for a given study region. No individuals were identified as American Indian/Alaska Native or Native Hawaiian/Pacific Islander.

Forty-nine percent of the studies used measures of greenness (e.g., normalized difference vegetation index), and 49% used measures of park/greenspace (Figure 3D). One study examined time spent in greenspace (2%) [58], eight included the distance to greenspace (14%) [27,64,66,67,74,80,91,93], and one used a measure of satisfaction in surrounding greenspaces (2%) [84]. Sixty-eight percent (*n* = 39) of the studies used cognitive test outcomes (e.g., global cognition, verbal intelligence, processing speed, etc.), 25% (*n* = 14) included ADRD diagnosis/incidence measures, 11% (*n* = 6) used neuroimaging measures (i.e., left hippocampal volume, white matter grade, etc.), and 4% (*n* = 2) used measures of subjective cognitive decline (Figure 3E). From the studies that evaluated greenness (e.g., normalized difference vegetation index), 39% of studies assessed associations with cognitive tests (*n* = 22), 7% with ADRD diagnosis (*n* = 4), and 7% with brain imaging (*n* = 4) (Figure 3F). From the studies that evaluated parks/greenspaces, 35% assessed associations with cognitive tests (*n* = 20), 16% with ADRD diagnosis (*n* = 9), and 4% with brain imaging measures (*n* = 2) (Figure 3F). Overall, 74% (*n* = 42) of the studies observed at least one positive association between greenspace and brain health, while 16% (*n* = 9) observed at least one negative association, and 81% (*n* = 46) observed at least one null association (Figure 4).

Only 4 of the 57 studies (7%) investigated differences in greenspace—brain health associations by the racialized/ethnic group (Table 2) [27,54,65,94]. The first was a longitudinal study using data from the population-based multiethnic study of atherosclerosis, which enrolled participants from six US locations: Forsyth County, North Carolina; New York, New York; Baltimore, Maryland; St. Paul, Minnesota; Chicago, Illinois; Los Angeles, California [27]. While the study found an association between greater park access and maintained/improved global cognition over time in the overall sample (versus declining over time) (OR = 1.04; 95% CI = 1.00–1.08; *p* = 0.04), a statistically significant difference was not observed by the racialized group when testing an interaction term in the multivariable models (i.e., park access×race, *p* = 0.85). However, a borderline association was observed for Black (OR: 1.07, 95% CI: 1.00, 1.14; *p* = 0.07) but not White participants (OR = 1.04, 95% CI: 0.96–1.12; *p* = 0.35) when stratifying the model by race. The second study used data on >100,000 U.S. Medicare patients (federal health insurance for ≥65-year-old Americans). In stratified analyses, the authors found no significant difference between Black and White patients in the association of the proportion of neighborhood greenspace and Alzheimer’s disease (AD) risk [94]. Greater greenspace was associated with reduced AD risk for both Black and White individuals. 

The third study, which used the CARTaGENE cohort from six metropolitan areas in Quebec, Canada, found no association between neighborhood greenness and three measures of cognition (reasoning, visual memory, and reaction time) in either the entire sample or sample stratified by racialized/ethnic groups (i.e., White and non-White) [54]. The last study included US Medicare beneficiaries aged >65 years living in Miami-Dade County, Florida, from 2010 to 2011 [65]. The authors observed that higher greenness was associated with reduced risk of AD, ADRD, and non-AD dementia, after adjusting for individual and neighborhood sociodemographic characteristics. To better understand the role of demographic variables in the relationship between greenness and AD, they conducted separate tests for the interaction between greenness and each of the demographic variables but found no significant interactions with racialized/ethnic groups.

## 4. Discussion

In this rapid review of 57 published studies, we found that the majority (74%) observed at least one positive association suggesting a benefit between greenspace exposure and brain health outcomes. Twenty-one percent of studies explicitly indicated that they included Black, Hispanic/Latinx, and/or Asian participants. However, it must be noted that several reviewed studies were conducted in countries that are predominantly composed of one of these racialized/ethnic groups (e.g., most participants in Chinese studies were likely Asian), and we did not presume and assign a racial/ethnic identity for the participants in those studies. We found that only 7% of the studies investigated differences in greenspace—brain health associations by the racialized/ethnic group, although minoritized racialized/ethnic groups are present in countries represented in the reviewed studies (e.g., Zhuang and Hui in China). Most studies were conducted in developed/high-income countries (81%). Lastly, none of the studies were framed by health disparities, social/structural determinants of health, or related frameworks. Overall, our rapid review highlights the significant lack of studies conducted in developing countries, specifically studies that examined differences in associations by the racialized/ethnic group and that were framed according to fundamental mechanisms (e.g., structural racism) that are responsible for the unequal distribution of greenspaces in our communities and that are hypothesized to relate to disparities in ADRD risk.

An identified gap in the published greenspace and brain health literature is the lack of attention to how greenspaces differentially affect ADRD risk factors by the racialized/ethnic group, as well as how ADRD risk factors differentially impact ADRD outcomes by the racialized/ethnic group (Figure 1). For instance, the availability of neighborhood parks may be more important for social interaction depending on the racialized/ethnic group [99]. Similarly, social interaction may be more important for preserving cognitive function in older age depending on the racialized/ethnic group [100]. Thus, new studies should better incorporate this type of upstream and downstream effect modification of the causal pathway between greenspaces and brain health outcomes. Table 3 lists other major methodological gaps to address in the future. The list has been devised to help guide subsequent studies that aim to be more inclusive of minoritized, racialized/ethnic groups and individuals from historically disadvantaged communities and lower- to middle-income countries and to increase awareness of underlying issues that likely will accompany studies of greenspace and brain health associations among minoritized, racialized/ethnic groups.

Health studies historically have had difficulty in recruiting diverse participants, and ADRD research is no exception. Results on greenspace—brain health associations from majority White cohorts and higher-income countries cannot be automatically extrapolated to minoritized groups and developing countries. 

To address the longstanding issue of diverse recruitment and retention, the US National Institute on Aging developed a guide that can be used to increase diversity in clinical studies of ADRD [102]. Recommended strategies that can be incorporated into brain health studies, such as those studying greenspace exposure/interventions, include developing partnerships and trust with underrepresented communities, promoting health and science literacy in the community, and increasing diversity in the research workforce to address bias. These strategies are particularly useful for new, prospective studies, whereas techniques for retention of underrepresented should be incorporated in preestablished cohort studies to minimize attrition (e.g., offering direct benefits whenever possible and researching topics directly pertinent to the community under study) [103].

Another methodological concern when examining greenspace—brain health associations by the racialized/ethnic group is the complex intertwining of neighborhood segregation, neighborhood SES, and racialized/ethnic group identity. These factors are difficult to disentangle given the historical roots of racism and residential segregation in the US and other countries [104,105]. Any studies of racialized/ethnic group differences in the neighborhood greenspace and brain health associations need to include sufficient samples of minoritized individuals from all neighborhood types (e.g., including lower segregation and higher SES neighborhoods) and/or should consider stratifying findings by these factors. Otherwise, any negative or null associations observed between greater access to greenspace and brain health outcomes among minoritized groups may be reflective of the underlying neighborhood conditions of segregated, low SES neighborhoods, which are often also more urban in character (e.g., with more crime, physical disorder, and fewer health-promoting resources and opportunities).

Inequities in S/SDOH, both proximal and distal, begin at birth and influence health outcomes across the life course [106]. Studies of dementia are often studies of survival. In the US, provisional estimates for 2021 show disparities in life expectancy by the racialized/ethnic group: 83.5 years for Asian individuals, 77.7 years for Hispanic/Latinx individuals, 76.4 years for White individuals, 70.8 years for Black individuals, and 65.2 years for Americans Indian/Alaskan Natives [107]. Age is the strongest risk factor for dementia, and the risk of ADRD is highest among those ages 65 and older [20]. Many minoritized individuals die before they can receive a diagnosis or are diagnosed at later stages of the disease [108]. This combined with a higher burden or comorbid disease and difficulty in continuously participating in research studies due to lack of time, transportation, or trust in the researchers can result in higher attrition of minoritized, racialized/ethnic groups in studies of brain health, including those focused on greenspace exposure. This review has identified greenspace as a potential protective factor for brain health, but more work is needed to measure how timing and duration of access to greenspace across the entire life impact survival (i.e., aging) and dementia risk and to ensure that minoritized groups remain enrolled in studies at similar rates as majoritized groups to avoid bias of findings.

A few studies have examined the relationship between park quality, access, and use by racialized/ethnic groups. Yet, perceptions of neighborhood quality, walkability, safety, and accessibility of green spaces, parks, and natural areas are known to affect physical activity among older adults [98,109]. Findings from previous studies suggest that in lower-income neighborhoods, proximity to greenspace and parks per se does not provide sufficient incentives for older adults to visit parks due to safety concerns and/or lack of amenities or adequate maintenance. Tinsley et al. (2002) found statistically significant differences among older adults from different racialized/ethnic groups in terms of intended park use, social and built environment characteristics, and perceived benefits [110]. Often parks in predominantly Black neighborhoods are perceived by the local residents as unattractive, unsafe, or exhibiting signs of disrepair [111]. Improvements in park quality in these neighborhoods as well as developing park programs and providing park fitness equipment for older adults are key interventions that can incentivize park usage by older adults [109,112]. Thus, future studies are needed to determine the importance of greenspace quality in promoting brain health throughout the life course, particularly for minoritized, racialized/ethnic communities.

Clinical evaluations for cognitive impairment and ADRD are based on healthcare systems and diagnostic tools that are often biased against minoritized, racialized groups. Black and other minoritized groups have been found to be at increased risk of underdiagnosis for dementia, which has been attributed to a multitude of factors, including but not limited to differences in dementia risk factors, disease etiology and presentation, cognitive test performance, and care-seeking behavior [113]. Undoubtedly, disparities in diagnosis are also related to inadequate access to healthcare, culturally/linguistically appropriate screening tools, and overarching structural racism, which suppresses opportunities for minoritized groups to receive appropriate evaluation and diagnosis [114]. Neuropsychological tests, which evaluate an individual’s cognitive functioning and decline over time and detect cognitive impairment and ADRD, are traditionally developed for majoritized populations, specifically White and English-speaking individuals. As such, numerous studies have demonstrated that non-White and non-English speaking individuals score consistently lower than majoritized groups on tests measuring various cognitive domains, necessitating different sets of norms for minoritized populations to accurately detect impairment [115]. Reasons for these disparities have been discussed in depth elsewhere, but they include structural factors such as differences in educational and occupational opportunity [115,116]. Disparities in cognitive testing and diagnosis need to be considered in any study of greenspace and ADRD outcomes that includes racialized/ethnic groups. Ideally, future studies will include cognitive tests that have been validated and/or back-translated for any racialized/ethnic group of interest, but at a minimum, care should be taken in the interpretation and potential bias of findings based on racialized/ethnic group disparities in testing and diagnosis.

Health studies on the built environment including greenspaces regularly note their limitations due to missing data on residential histories and participants’ reasons for moving to neighborhoods, which ultimately may bias study findings [117]. Accumulated exposure to greenspace environments over the life course may be a better predictor of late-life cognition than greenspace exposure in later life. In addition, controlling for early and midlife neighborhood conditions, such as neighborhood SES, is important when trying to understand the independent influence of neighborhood greenspace on late-life brain health. Exposure to environments with greater or fewer greenspaces may have accumulated impacts on individuals, and the differences may be particularly stark when comparing minoritized, racialized/ethnic groups with majoritized groups. The characterization of neighborhood environments throughout the life course is not possible without detailed residential history data (e.g., using life grid methods [26] or other questionnaires [118,119]), or at a minimum, self-reported measures of neighborhood environments over time. Yet few studies have incorporated such data collection to date. Similarly, most studies do not collect information on reasons for living in and moving to certain neighborhoods, particularly in late life. Self-selection into neighborhoods can result in reverse causation (e.g., harbingers of dementia prompt a move from an urban area to an assisted living facility in a suburban locale that is greener). However, unlike majoritized groups who have greater neighborhood choices, self-selection is a misnomer for minoritized groups, who have restricted choices in the neighborhoods in which they can live. Future studies will need to adequately consider these factors and their potential influence on study findings regarding differences by racialized/ethnic groups.

Lastly, researchers have called for measures of spatial polygamy, which is the “simultaneous belonging or exposure to multiple nested and non-nested, social and geographic, real, virtual and fictional, and past and present contexts” [101]. In this review, a single study assessed time spent in greenspace [58]; however, none captured multifaceted environmental exposures, including greenspace, as proposed by spatial polygamy. A motivation of spatial polygamy is to avoid exposure misclassification by singling out one element in our environments in a particular place (e.g., neighborhood) or time and confounding by factors closely related to the environment characteristic of interest (e.g., greenspaces and air pollution exposure). While studies would benefit from an eye toward spatial polygamy, tradeoffs must be considered (e.g., selection bias and generalizability to diverse populations) when asking for multiple detailed questionnaires/diaries over time or consent to use of mobility/exposure tracking devices or apps over multiple periods, which may deter minoritized, racialized/ethnic groups due to factors such as mistrust/privacy concerns and burden [120,121,122].

The primary limitation of this study is that it is a rapid review. Systematic reviews employ more rigorous methods to provide a comprehensive review of published literature, sometimes additionally including grey literature, risk of bias, and meta-analyses to estimate effect size. This study’s sample size of 57 papers may not be representative of the entire body of literature on the topic. Despite the benefits of a systematic review, a rapid review served our purpose to estimate the extent to which the published literature has included diverse individuals/regions and has examined associations by the racialized/ethnic group, and provided avenues for future research on greenspace and brain health among minoritized, historically disadvantaged, and developing country populations. Another limitation of this study is that many reviewed studies (72%) did not explicitly specify the racialized/ethnic groups included, particularly among studies conducted in international settings. For studies conducted in regions with at least moderate diversity in their racialized/ethnic groups (e.g., Ontario), this lack of attention to racialized/ethnic group identity of their participants highlights a lack of sensitivity to diversity, either through the absence of a description, lack of pertinent data, of or lack of inclusion of minoritized individuals. A lack of racial/ethnic and geographic diversity has been described in other medical research fields, particularly regarding participant inclusion in clinical trials [123]. In this rapid review, we observed a dearth of research on greenspace and brain health guided by health disparities, social/structural determinants of health, or related frameworks, despite the known differences in both greenspace availability/quality and dementia risk by the racialized/ethnic group and geography. Our findings imply that non-representativeness of participants from diverse racial and ethnic groups in study cohorts or geographic regions obstructs the external validity of the majority of current observational dementia research [124]. This has important implications for future researchers that seek to examine greenspace exposure as a potential population-level approach to improve cognitive health and suggests the need to more carefully consider underrepresented groups and regions and incorporate health disparities frameworks in future work.

## 5. Conclusions

Overall, this study demonstrated a paucity of greenspace—brain health research from developing countries, studies focused on differences in associations and differences in causal mechanisms between racialized/ethnic groups, and studies purposefully focused on associations within a specific minoritized, racialized/ethnic group. These scientific gaps should be addressed to ensure that greenspace interventions, policies, and plans to improve well-being and brain health will truly benefit historically disadvantaged communities. For instance, while in theory, new and improved parks in disadvantaged communities should improve park use and, thereby, the health of residents, it may have the opposite effect by gentrifying neighborhoods, which can decrease long-term residents’ sense of belonging and park use compared with incoming residents [125]. Despite the potential negative benefits, disadvantaged communities have the right to equal access to health-promoting greenspaces, which in several studies have been tied to health benefits for minoritized groups [126]. Ultimately, affordable housing and related policies may be required to be implemented in tandem with greenspace interventions to counteract any potentially negative impacts on historically disadvantaged communities [127].

## Figures and Tables

**Figure 1 ijerph-20-05666-f001:**
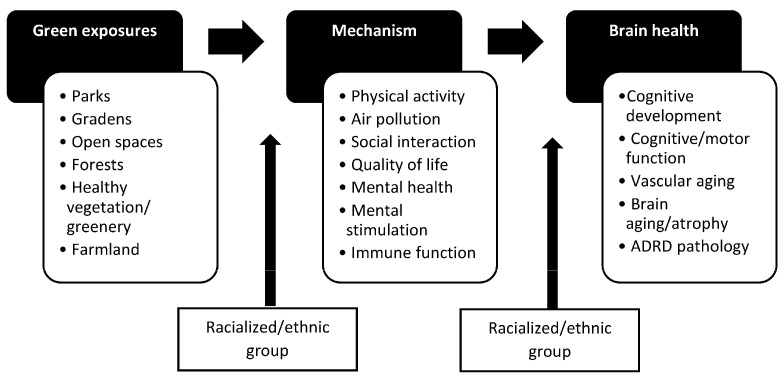
Conceptual framework for greenspace and brain health associations and effect modification by the racialized/ethnic group.

**Figure 2 ijerph-20-05666-f002:**
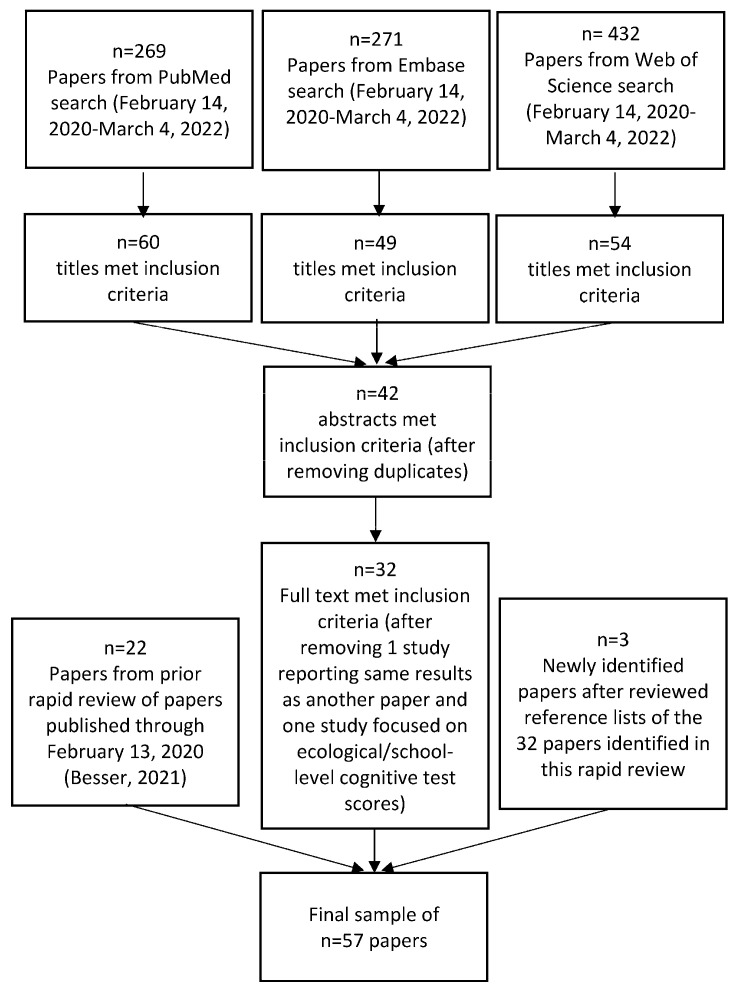
Sample size flow diagram for rapid literature review.

**Figure 3 ijerph-20-05666-f003:**
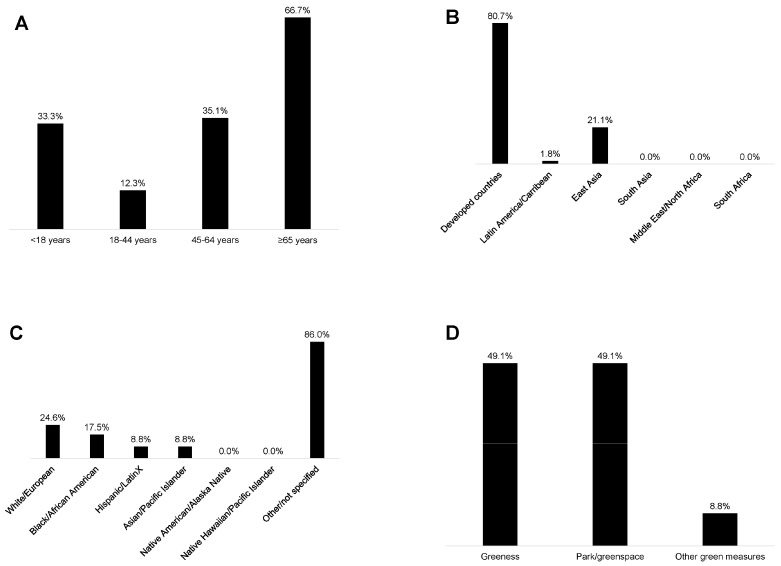
Studies in the rapid literature review by (**A**) participants’ age group, (**B**) geographic region, (**C**) participants’ racialized/ethnic group, (**D**) green exposure studied, (**E**) brain health outcome studied, and (**F**) associations examined (not mutually exclusive categories).

**Figure 4 ijerph-20-05666-f004:**
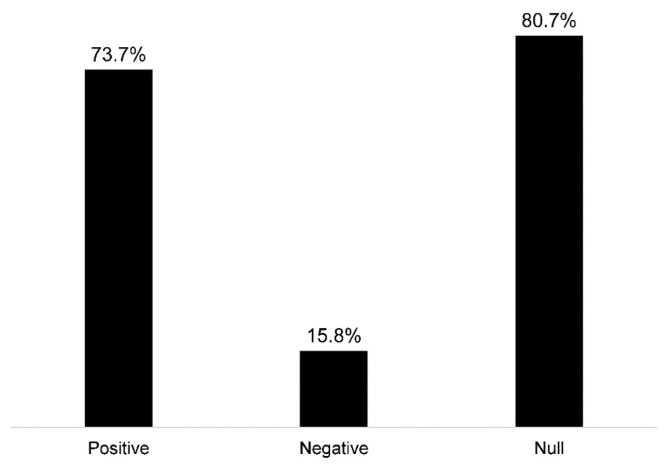
Observed associations among reviewed studies between greenspace and ADRD brain health outcomes (not mutually exclusive categories).

**Table 1 ijerph-20-05666-t001:** Summary of rapid literature review findings by geographic location and racialized/ethnic group.

First Author (Year)	Sample Size	Age Groups Included	Geographic Location	The Study Included the Following Racialized/Ethnic Group(s) **					Association
				White/European	Black/ African American	Hispanic/ LatinX	Asian Pacific Islander	Did Not Specify Any	
Aitken (2021) (CS) [65]	249,405	≥65 years	US	Yes	Yes	Yes	No	No	Grness-Dx: + Grness-Dx: N
Almeida (2022) (CS) [66]	3827	<18 years	Portugal	No	No	No	No	Yes	GrSp-Cog: + GrSp-Cog: N Grness-Cog: − Grness-Cog: N
Anabitarte (2022) (LC) [67]	751	<18 years	Spain	No	No	No	No	Yes	Grness-Cog: + Grness-Cog: N GrSp-Cog: N
Asta (2021) (CS) [68]	465	<18 years	Italy	No	No	No	No	Yes	Grness-Cog: + Grness-Cog: N
Astell-Burt (2020) (LC) [9]	45,644	45–64 years, ≥65 years	Australia	No	No	No	No	Yes	OthGr-SubCog: + OthGr-SubCog: N
Astell-Burt (2020) (LC) [70]	109,688	45–64 years, ≥65 years	Australia	No	No	No	No	Yes	GrSp-Dx: − GrSp-Dx: + GrSp-Dx: N
Bagheri (2021) (CS) [96]	25,511	≥65 years	Australia	No	No	No	No	Yes	GrSp-Dx: + GrSp-Dx: N
Besser (2020) (CS) [71]	4084	45–64 years, ≥65 years	US	Yes	Yes	Yes	Yes	No	GrSp-Cog: + GrSp-Cog: − GrSp-Cog: N
Besser (2021) (LC) [27]	1733	45–64 years, ≥65 years	US	Yes	Yes	Yes	Yes	No	GrSp-Cog: + GrSp-Cog: N
Besser (2021) (CS) [72]	1125	≥65 years	US	Yes	Yes	No	No	Yes	GrSp-Img: N
Bijnens (2020) (CS) [73]	620	<18 years	Belgium	No	No	No	No	Yes	GrSp-Cog: + GrSp-Cog: N
Bijnens (2022) (CS) [74]	596	<18 years	Belgium	No	No	No	No	Yes	GrSp-Cog: + GrSp-Cog: N
Binter (2022) (CS) [75]	5403	<18 years	UK, France, Spain, Greece	No	No	No	No	Yes	Grness-Cog: + Grness-Cog: N GrSp-Cog: N
Brown (2018) (CS) [47]	249,405	≥65 years	US	Yes	Yes	Yes	No	No	Grness-Dx: +
Cerin (2021) (CS) [76]	4141	18–44 years, 45–64 years, ≥65 years	Australia	No	No	No	No	Yes	GrSp-Cog: +
Cherrie (2018) (LC) [26]	281	<18 years, 18–44 years, 45–64 years, ≥65 years	UK	No	No	No	No	Yes	GrSp-Cog: + GrSp-Cog: N
Cherrie (2019) (LC) [48]	281	≥65 years	UK	No	No	No	No	Yes	GrSp-Cog: + GrSp-Cog: N
Clarke (2012) (CS) [49]	949	45–64 years, ≥65 years	US	Yes	Yes	Yes	No	No	GrSp-Cog: N
Crous-Bou (2020) (CS) [77]	958	45–64 years, ≥65 years	Spain	No	No	No	No	Yes	Grness-Cog: N Grness-Img: + Grness-Img: N
Dadvand (2015) (LC) [50]	2593	<18 years	Spain	No	No	No	No	Yes	Grness-Cog: + Grness-Cog: N
Dadvand (2017) (LC) [51]	987	<18 years	Spain	No	No	No	No	Yes	Grness-Cog: + Grness-Cog: N
Dadvand (2018) (CS) [30]	253	<18 years	Spain	No	No	No	No	Yes	Grness-Img: + Grness-Img: N
De Keijzer (2018) (LC) [29]	6506	45–64 years, ≥65 years	UK	No	No	No	No	Yes	Grness-Cog: + Grness-Cog: N
Dockx (2022) (CS) [78]	456	<18 years	Belgium	No	No	No	No	Yes	GrSp-Cog: + GrSp-Cog: N
Dzhambov (2019) (CS) [52]	112	<18 years	Bulgaria	No	No	No	No	Yes	Grness-Cog: + Grness-Cog: N Grness-Img: + Grness-Img: N
Falcon (2021) (CS) [79]	212	45–64 years, ≥65 years	Spain	No	No	No	No	Yes	Grness-Img: + Grness-Img: N
FangFang (2022) (CS) [80]	5848	45–64 years, ≥65 years	China	No	No	No	No	Yes	GrSp-Cog: N
Finlay (2021) (CS) [81]	21,151 (quantitative) 125 (qualitative)	45–64 years, ≥65 years	US	Yes	Yes	No	No	Yes	GrSp-Cog: +
Flouri (2019) (CS) [53]	4758	<18 years	UK	No	No	No	No	Yes	GrSp-Cog: +
Hystad (2019) (CS) [54]	6658	18–44 years, 45–64 years, ≥65 years	Canada	No	No	No	No	Yes	Grness-Cog: − Grness-Cog: N
Jimenez (2022) (CS) [82]	857	<18 years	US	Yes	No	No	No	Yes	Grness-Cog: + Grness-Cog: N
Jin (2021) (CS) [83]	1199	≥65 years	China	No	No	No	Yes	No	Grness-Cog: + Grness-Cog: N
Ju (2021) (CS) [84]	191,054	18–44 years, 45–64 years, ≥65 years	Korea	No	No	No	No	Yes	OthGr-SubCog: N
Julvez (2021) (CS) [85]	1298	<18 years	UK, France, Spain, Lithuania, Norway, Greece	No	No	No	No	Yes	Grness-Cog: + Grness-Cog: − Grness-Cog: N
Kuhn (2017) (CS) [31]	341	45–64 years, ≥65 years	Germany	No	No	No	No	Yes	GrSp-Img: + GrSp-Img: N
Lee (2021) (CS) [86]	189	<18 years	Korea	No	No	No	No	Yes	Grness-Cog: + Grness-Cog: N
Lega (2021) (CS) [87]	185	18–44 years, 45–64 years, ≥65 years	UK	No	No	No	No	Yes	Grness-Cog: + Grness-Cog: N
Liao (2019) (CS) [55]	1312	<18 years	China	No	No	No	No	Yes	Grness-Cog: +
Liu (2019) (LC) [88]	24,802	≥65 years	Taiwan	No	No	No	No	Yes	GrSp-dx: N
Liu (2020) (LC) [89]	52,412	≥65 years	Taiwan	No	No	No	No	Yes	GrSp-dx: N
Maes (2021) (LC) [97]	3568	<18 years	UK	Yes	Yes	No	Yes	Yes	GrSp-Cog: + GrSp-Cog: N
Paul (2020) (LC) [90]	1.74 million dementia cohort, 4.25 million stroke cohort	18–44 years, 45–64 years, ≥65 years	Canada	No	No	No	No	Yes	Grness-Dx: +
Reuben (2019) (LC) [56]	1658	<18 years	UK	No	No	No	No	Yes	Grness-Cog: N
Slawsky (2022) (LC) [91]	3047	≥65 years	US	Yes	No	No	No	Yes	OthGr-Dx: +
Sylvers (2022) (CS) [98]	10,289	45–64 years, ≥65 years	US	Yes	Yes	No	No	No	GrSp-Cog: + GrSp-Cog: N
Tani (2021) (LC) [92]	76,053	≥65 years	Japan	No	No	No	No	Yes	GrSp-Dx: N
Wang (2017 (CS) [57]	3544	≥65 years	China	No	No	No	No	Yes	Grness-Cog: N
Ward (2016) (CS) [58]	72	<18 years	New Zealand	No	No	No	No	Yes	OthGr-Cog: N
Wu (2015) (CS) [60]	2424	≥65 years	UK	No	No	No	No	Yes	GrSp-Cog: − GrSp-Dx: −
Wu (2017) (CS) [59]	7505	≥65 years	UK	No	No	No	No	Yes	GrSp-Cog: − GrSp-Dx: N
Wu (2020) (CS) [93]	CFAS: 4955 10/66: 3386	≥65 years	UK China, Dominican Republic, Mexico	No	No	No	No	Yes	GrSp-Dx: − GrSp-Dx: N
Wu (2021) (LC) [94]	~106,763	≥65 years	US	Yes	Yes	No	No	No	GrSp-Dx: +
Yu (2018) (CS) [61]	3240	≥65 years	China	No	No	No	No	Yes	Grness-Cog: N
Yuchi (2020) (LC) [62]	678,000	45–64 years, ≥65 years	Canada	No	No	No	No	Yes	Grness-Dx: + Grness-Dx: −
Zhu (2019) (LC) [95]	38,327	≥65 years	China	No	No	No	No	Yes	Grness-Cog: + Grness-Cog: N
Zhu (2020) (LC) [63]	6994	≥65 years	China	No	No	No	Yes	No	Grness-Cog: +
Zijlema (2017) (CS) [64]	1628	18–44 years, 45–64 years, ≥65 years	Spain, Netherlands, UK	No	No	No	No	Yes	GrSp-Cog: + GrSp-Cog: N Grness-Cog: N OthGr-Cog: N
Total studies by racialized/ethnic group				14	10	5	5	41	--
Studies with positive associations									42
Studies with inverse associations									9
Studies with null associations									46 (11 had only null associations)
Total									57 studies

Abbreviations: +, positive association; −, inverse association; N, null association; Grness = greenness; GrSp = greenspace/park space (amount/area or percentage of any greenspace type (park/forest/urban green/woodland/etc.) or distance to nearest greenspace); OthGr = other green measures (time spent in greenspace, visits to greenspace, greenspace quality); Dx = ADRD diagnosis; Cog = cognition; Img = brain imaging; SubCog = subjective cognitive decline; CS = cross-sectional outcome; LC = included longitudinal outcome measuring change over time (e.g., the incidence of diagnosis or change in cognition over time); ** No studies including Native American/Alaska Native or Native Hawaiian/Pacific Islander individuals.

**Table 2 ijerph-20-05666-t002:** Four of fifty-seven reviewed studies examined differences in greenspace—brain health associations by the racialized/ethnic group.

Citation	Method	Finding
Aitken et al. (2021) [65]	Interaction term testing (e.g., NDVI x racialized/ethnic group)	No interaction was indicated between NDVI and racialized/ethnic group (Non-Hispanic White, Hispanic, Black) in relation to the odds of ADRD diagnosis (results not reported).
Besser et al. (2021) [27]	Stratification by racialized/ethnic group and interaction term testing (i.e., percentage park space x racialized/ethnic group)	Among Black participants, neighborhood percentage park space was borderline associated with maintained/improved global cognition. No association for White participants. The interaction term was not statistically significant.
Wu & Jackson (2021) [94]	Stratification by racialized/ethnic group	Neighborhood percentage greenspace associated with reduced Alzheimer’s risk in both Black and White individuals. No difference in association based on racialized/ethnic group.
Hystad et al. (2019) [54]	Stratification by racialized/ethnic group	No difference in the association between neighborhood greenness (NDVI) and cognitive functioning (reasoning, visual memory, and reaction time) between White and non-White participants.

**Table 3 ijerph-20-05666-t003:** Methodological concerns in examining differences in greenspace—brain health associations by the racialized/ethnic group.

Concern/Topic	Details
Segregation of neighborhoods by racialized/ethnic group	Residential neighborhood correlated with individual’s racialized/ethnic group, amount and type/quality of neighborhood greenspace, and individual and neighborhood SES. Difficult to disentangle these factors.
Insufficient recruitment of diverse cohorts	Lack of focus on recruiting diverse samples from the start (not a priority). Difficulty in recruiting diverse samples due to lack of experience/knowledge/effort by investigators, lack of trust of investigators, and lack of diverse researchers/study team.
Differential attrition by racialized/ethnic group	Racialized/ethnic groups are often less likely to enroll in studies and more likely to drop out over time due to social determinants.
Overemphasis on statistical significance and comparison with the White reference group (e.g., testing differences using interaction terms)	Heterogeneity within racialized/ethnic groups is often ignored with a preference for a comparison group.
Differences in greenspace exposure and quality by racialized/ethnic group	Measuring the amount of neighborhood greenspace/greenness may obscure differences in true exposure to greenspace or quality that may significantly differ by racialized/ethnic group (e.g., differences in preference for greenspaces or time spent in neighborhoods due to differences in transportation modes)
Neuropsychological tests developed primarily for the majority (e.g., White individuals)	Inadequate accounting for cultural differences that affect testing/scores and historical, upstream factors that affect cognitive test scores differential by racialized/ethnic group.
Differences in residential moves and neighborhood self-selection by racialized/ethnic group	Important to consider the context of residential moves and how it likely differs depending on racialized/ethnic group (e.g., out of choice or necessity). Many studies emphasize potential bias from self-selection into greener neighborhoods that promote physical activity, but this may not be as common for minoritized groups.
Challenges in obtaining detailed or technology-based measures of the complex, multi-faceted environmental exposures (e.g., spatial polygamy [101])	Studies requiring long-term, detailed, or technologically intensive data acquisition can be a deterrent to participation for minoritized racialized/ethnic groups.

## Data Availability

As a review of published papers, this study has no accompanying data.

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
