# Peer review of "Diversity of Studies on Neighborhood Greenspace and Brain Health by Racialized/Ethnic Group and Geographic Region: A Rapid Review"

_ijerph, 2023, doi:10.3390/ijerph20095666_

Round 1

Reviewer 1 Report

The rapid review by Besser et al. is an interesting report. However, additional modifications might be needed.

1. The structure of the manuscript is not clear—is it a review or is it a research manuscript? Reorganizing the report might be needed.

2. Either way, the questions the authors are trying to answer are not clear.

Is it the effect of living in or having access to a green area on the risk of AD among minority racial groups?

If that is the case, do the authors expect to study the effect of green spaces on African minorities in China?

It is well known that there might not be a significant African minority in China.

China minorties inlcude the Ugur.  Minorities in the United Kingdom include Asians, Africans, and others. In Italy, migrants are predominantly from Africa. Thus, if the authors decide to pursue specifically the question of minorities, then more attention might be needed to the nature of minorities in each country.

The study or review might need additional statistical analyses that really pinpoint the answers.

Although more and higher quality figures might be needed,

Additionally, clarity in the abstract and the discussion section about the findings might be necessary.

Author Response

RESPONSE TO REVIEWERS

Reviewer 1:

The rapid review by Besser et al. is an interesting report. However, additional modifications might be needed.

Response: Thank you for taking the time to review our manuscript and for your comments.

  1. The structure of the manuscript is not clear—is it a review or is it a research manuscript? Reorganizing the report might be needed.

Response: The manuscript is a rapid literature review, which is indicated in the title (i.e., Title: “Diversity of Studies on Neighborhood Greenspace and Brain Health by Racialized/Ethnic Group and Geographic Region: A Rapid Review”), abstract, aims, and methods.

For instance, in the last paragraph of the introduction (line 123), we state: “Thus, in this study, we aimed to determine the extent to which published green-space-brain health studies have examined different racial/ethnic groups and geographic regions. We conducted a rapid literature review [42] to delineate the diversity of published greenspace-brain health studies with respect to included racialized/ethnic groups and geographic regions, the extent that published studies have investigated racialized/ethnic group differences in associations, and methodological issues surrounding studies of racialized/ethnic group disparities in greenspace and brain health associations.”

We are happy to reorganize or add additional structure if you have specific suggestions. However, we believe the paper structure that we employed, the tables and figures that we presented, and the emphasis that it is a rapid literature review throughout the various sections of the paper helps indicate it is a review and not a research manuscript. However, to help clarify that our presented findings are from a literature review, we have updated the titles of the tables and figures as follows:

  • Figure 1 changed to “Sample size flow diagram for rapid literature review”
  • Table 1 changed to “Summary of rapid literature review findings by geographic location and racialized/ethnic group”
  • Table 2 changed to “Four of 57 reviewed studies examined differences in greenspace-brain health associations by racialized/ethnic group”
  • Figure 3 changed to “Studies in rapid literature review by A) participants’ age group, B) geographic region, C) participants’ racialized/ethnic group, D) green exposure studied, E) brain health outcome studied; and F) associations examined (not mutually exclusive categories)”
  • Figure 4 changed to “Observed associations among reviewed studies between greenspace and ADRD brain health outcomes”
  1. Either way, the questions the authors are trying to answer are not clear.

Response: In the abstract (line 26), we state: “In this rapid literature review, we: 1) describe the diversity of published greenspace-brain health studies with respect to racialized/ethnic groups and geographic regions, 2) determine the extent published studies have investigated racialized/ethnic group differences in associations, and 3) review methodological issues surrounding studies of racialized/ethnic group disparities in greenspace and brain health associations.”

Also, in the introduction section (line 125), we state: “We conducted a rapid literature review [42] to delineate the diversity of published greenspace-brain health studies with respect to included racialized/ethnic groups and geographic regions, the extent that published studies have investigated racialized/ethnic group differences in associations, and methodological issues surrounding studies of racialized/ethnic group disparities in greenspace and brain health associations.”

The findings related to these specific aims are then reported in the results (aim 1 and 2) and discussion sections (aim 3).

To further address your concern, we have added another sentence in the last paragraph of the introduction to further clarify the research question (line 123): “Thus, in this study, we aimed determine the extent to which published greenspace-brain health studies have examined different racial/ethnic groups and geographic regions.”

  1. Is it the effect of living in or having access to a green area on the risk of AD among minority racial groups? If that is the case, do the authors expect to study the effect of green spaces on African minorities in China?

Response: Apologies for any confusion, but we did not aim to assess the effect of living in or having access to a green space on the risk of AD among minority racial groups. Instead, we aimed to describe the diversity of published greenspace-brain health studies with respect to racialized/ethnic groups and geographic regions included. To our knowledge, no literature reviews on greenspace and AD outcomes have summarized the racialized/ethnic group and geographic variation of published studies as is done in our rapid literature review. We hope our response to your comment directly above helps clarify the aims of our study.

  1. It is well known that there might not be a significant African minority in China. China minorties inlcude the Ugur.  Minorities in the United Kingdom include Asians, Africans, and others. In Italy, migrants are predominantly from Africa. Thus, if the authors decide to pursue specifically the question of minorities, then more attention might be needed to the nature of minorities in each country.

Response: Please see our reply to your comments directly above. We did not aim to study the effect of green spaces on minority groups but to summarize the extent that research done to date has included different racial/ethnic groups and been conducted in different regions of the world. However, we have now added into the first paragraph of the discussion the following to provide a more nuanced discussion of minorities in countries such as China (see line 286):

“Twenty-one percent of studies explicitly indicated that they included Black, Hispanic/Latinx, and/or Asian participants. However, it must be noted that several reviewed studies were conducted in countries that are predominantly comprised of one of these racialized/ethnic groups (e.g., most participants in Chinese studies are likely Asian), but we did not presume and assign a racial/ethnic identity for the participants in those studies. We found that only 7% of the studies investigated differences in greenspace-brain health associations by racialized/ethnic group, although minoritized racialized/ethnic groups are present in countries represented in the reviewed studies (e.g., Zhuang and Hui in China).

  1. The study or review might need additional statistical analyses that really pinpoint the answers.

Response: Since the goal of the study was to describe the extent that published studies have included various racial/ethnic groups and geographic regions, we did not perform additional analyses such as a meta-analysis of results from the studies combined, which would be inappropriate based on the study aims (i.e., we were not trying to determine the strength of the evidence for associations between greenspace and brain health outcomes across racial/ethnic groups or geographic regions). Instead, consistent with our aims, we describe basic characteristics of the studies conducted to date (e.g., geographic regions, racial/ethnic groups, green space exposure used), as seen in Figure 3 for example.

  1. Although more and higher quality figures might be needed.

Response: Thank you for catching this. We have replaced Figure 3 to include higher resolution images. (The other figures remain clear when zoomed in.)

  1.  Additionally, clarity in the abstract and the discussion section about the findings might be necessary.

Response: In the abstract and in the discussion, we report the major findings of the literature review as presented in the results section, tables, and figures.

In the abstract (line 31), we state: “Of the 57 papers meeting our inclusion criteria as of March 4, 2022, 21% (n=12) explicitly identified and included individuals who were Black, Hispanic/Latinx, and/or Asian. Twenty-one percent of studies (n=12) were conducted in developing countries (e.g., China, Dominican Republic, Mexico), and 7% (n=4) examined racialized/ethnic group differences in greenspace-brain health associations. None of the studies were framed by health disparities, social/structural determinants of health, or related frameworks, despite the known differences in both greenspace availability/quality and dementia risk by racialized/ethnic group and geography”.

In the first paragraph of the discussion (line 284), we state: “In this rapid review of 57 published studies, we found that the majority (74%) observed at least one positive association suggesting a benefit between greenspace exposure and brain health outcomes. Twenty-one percent of studies explicitly indicated that they included Black, Hispanic/Latinx, and/or Asian participants. However, it must be noted that several reviewed studies were conducted in countries that are predominantly comprised of one of these racialized/ethnic groups (e.g., most participants in Chinese studies are likely Asian), but we did not presume and assign a racial/ethnic identity for the participants in those studies. We found that only 7% of the studies investigated differences in greenspace-brain health associations by racialized/ethnic group, although minoritized racialized/ethnic groups are present in countries represented in the reviewed studies (e.g., Zhuang and Hui in China). Most studies were conducted in developed/high-income countries (81%). Lastly, none of the studies were framed by health disparities, social/structural determinants of health, or related frameworks.  Overall, our rapid review highlights the significant lack of studies conducted in developing countries, that examined differences in associations by racialized/ethnic group, and that were framed according to fundamental mechanisms (e.g., structural racism) that are responsible for the unequal distribution of greenspaces in our communities that are hypothesized to relate to disparities in ADRD risk.”

We are happy to address any issues of clarity in the abstract and discussion section if you can provide more specifics on what needs clarification. 

Reviewer 2 Report

The manuscript by the authors addresses an important gap in the literature by summarizing the racialized/ethnic group and geographic variation in the published studies examining the association between greenspace and ADRD outcomes. The authors' systematic approach in describing the diversity of published greenspace-brain health studies with respect to racialized/ethnic groups and geographic regions is well-organized and provides valuable insights into the current state of the field.

However, the study's sample size is limited to 57 papers meeting the inclusion criteria as of March 4, 2022. While this may be understandable given the rapid rise of studies on this topic, the limited sample size may not be representative of the entire body of literature on the topic. Additionally, the authors acknowledge that none of the studies were framed by health disparities, social/structural determinants of health, or related frameworks, despite the known differences in both greenspace availability/quality and dementia risk by racialized/ethnic group and geography. It would have been useful for the authors to discuss the implications of this limitation in more detail and its potential impact on the interpretation of the findings.

Despite these limitations, the manuscript provides a valuable contribution to the literature on greenspace and brain health. The authors' conclusion that studies are needed in developing countries and that directly investigate racialized/ethnic group disparities in greenspace-brain health associations to promote health equity is an important recommendation for future research.

Overall, the manuscript is well-written, and the authors' approach and findings are significant. The manuscript is recommended for publication after addressing the limitations and providing more context regarding the implications of the findings. Future research should aim to address the gaps in the literature highlighted by the authors, including the need for a more comprehensive understanding of the relationship between greenspace and ADRD outcomes in different racialized/ethnic groups and geographic regions.

Author Response

RESPONSE TO REVIEWERS

Reviewer 2:

The manuscript by the authors addresses an important gap in the literature by summarizing the racialized/ethnic group and geographic variation in the published studies examining the association between greenspace and ADRD outcomes. The authors' systematic approach in describing the diversity of published greenspace-brain health studies with respect to racialized/ethnic groups and geographic regions is well-organized and provides valuable insights into the current state of the field.

Response: Thank you for your review and thoughtful comments.

  1. However, the study's sample size is limited to 57 papers meeting the inclusion criteria as of March 4, 2022. While this may be understandable given the rapid rise of studies on this topic, the limited sample size may not be representative of the entire body of literature on the topic. Additionally, the authors acknowledge that none of the studies were framed by health disparities, social/structural determinants of health, or related frameworks, despite the known differences in both greenspace availability/quality and dementia risk by racialized/ethnic group and geography. It would have been useful for the authors to discuss the implications of this limitation in more detail and its potential impact on the interpretation of the findings.

Response: We agree with your comment on the limitations of the sample size and have added a brief statement in the discussion section. In addition, we have expanded the discussion on the implications of the lack of research framed by health disparities, social/structural determinants of health or related frameworks.

Manuscript change (lines 441): “This study's sample size of 57 papers may not be representative of the entire body of literature on the topic.”

Manuscript change (line 453): “A lack of racial/ethnic and geographic diversity has been described in other medical research fields, particularly regarding participant inclusion in clinical trials. In this rapid review, we observed a dearth of research on green space and brain health guided by a health disparities, social/structural determinants of health, or related frameworks, despite the known differences in both greenspace availability/quality and dementia risk by racialized/ethnic group and geography. Our findings imply that non-representativeness of participants from diverse racial and ethnic groups in study cohorts or geographic regions obstructs the external validity of the majority of current observational dementia research. This has important implications for future researchers that seek to examine greenspace exposure as a potential population-level approach to improve cognitive health and suggests the need to more carefully consider underrepresented groups and regions and incorporate health disparities frameworks in future work.”

  1. Despite these limitations, the manuscript provides a valuable contribution to the literature on greenspace and brain health. The authors' conclusion that studies are needed in developing countries and that directly investigate racialized/ethnic group disparities in greenspace-brain health associations to promote health equity is an important recommendation for future research.

Response: Thank you very much. We appreciate your shared interest.

  1. Overall, the manuscript is well-written, and the authors' approach and findings are significant. The manuscript is recommended for publication after addressing the limitations and providing more context regarding the implications of the findings. Future research should aim to address the gaps in the literature highlighted by the authors, including the need for a more comprehensive understanding of the relationship between greenspace and ADRD outcomes in different racialized/ethnic groups and geographic regions.

Response: We believe that your comments helped improve the quality of our manuscript and we appreciate it.

Round 2

Reviewer 1 Report

Dear authors,

Thank you for addressing my concerns.

Kind regards.